# The Relationship between Metabolically Healthy Obesity and the Risk of Cardiovascular Disease: A Systematic Review and Meta-Analysis

**DOI:** 10.3390/jcm8081228

**Published:** 2019-08-15

**Authors:** Tzu-Lin Yeh, Hsin-Hao Chen, Szu-Ying Tsai, Chien-Yu Lin, Shu-Jung Liu, Kuo-Liong Chien

**Affiliations:** 1Department of Family Medicine, Hsinchu MacKay Memorial Hospital, No. 690, Section 2, Guangfu Road, East District, Hsinchu City 30071, Taiwan; 2Institute of Epidemiology and Preventive Medicine, National Taiwan University, No.17, Xu-Zhou Rd., Taipei City 10055, Taiwan; 3MacKay Junior College of Medicine, Nursing, and Management, No. 92, Shengjing Rd., Beitou Dist., Taipei City 11260, Taiwan; 4Department of Medicine, MacKay Medical College, No.46, Sec. 3, Zhongzheng Rd., Sanzhi Dist., New Taipei City 25245, Taiwan; 5Department of Family Medicine, MacKay Memorial Hospital, No. 92, Section 2, Zhongshan North Road, Taipei City 10449, Taiwan; 6Department of Pediatrics, Hsinchu MacKay Memorial Hospital, No. 690, Section 2, Guangfu Road, East District, Hsinchu City 30071, Taiwan; 7Department of Medical Library, MacKay Memorial Hospital, Tamsui Branch, No. 45, Minsheng Road, Tamsui District, New Taipei City 25160, Taiwan; 8Department of Internal Medicine, National Taiwan University Hospital, No. 7, Zhongshan S. Rd., Zhongzheng Dist., Taipei City 10002, Taiwan

**Keywords:** metabolically healthy obesity, cardiovascular disease, all-cause mortality, meta-analysis

## Abstract

Cardiovascular disease (CVD) risk in individuals with metabolically healthy obesity (MHO) is unclear. We searched databases from inception to May 2019. Data were pooled using a random effects model. Newcastle-Ottawa Scale assessment was performed. Primary and secondary outcomes were CVD risk and all-cause mortality. Forty-three studies involving 4,822,205 cases were included. The median percentage of females, age and duration of follow-up was 52%, 49.9 years and 10.6 years, respectively. The mean Newcastle-Ottawa Scale score of the articles was 7.9 ± 1.0. Compared to individuals with a metabolically healthy normal weight, individuals with MHO had higher adjusted risk of CVD and all-cause mortality. We identified a significant linear dose-response relationship between body mass index (BMI) and CVD risk among metabolically healthy individuals (*p* < 0.001); every unit increase in BMI increased the CVD risk. Multivariate meta-regression analysis showed that an increased proportion of women and age resulted in the risk of CVD affected by MHO reduction (*p* = 0.014, *p* = 0.030, respectively). Age and sex explained the observed heterogeneity and reported the adjusted *R*^2^. MHO resulted in a significantly increased risk for CVD; therefore, long-term weight loss should be encouraged.

## 1. Introduction

According to the World Health Organization, cardiovascular disease (CVD) is the leading cause of mortality worldwide, with a rate of 17.7 million deaths per year, which continues to increase every year [1]. Based on the Global Burden of Disease Study, the overall global cardiovascular mortality rate increased by nearly 41% between 1990 and 2013 [2]. Obesity is an independent risk factor for CVD and all-cause mortality [3,4,5,6] due to the various physiological and metabolic changes that are associated with the condition. However, the risk for CVD differs between different obesity phenotypes [7,8].

Obesity is not a uniform condition [8,9,10]. Generally, obesity is categorized into four phenotypes according to metabolic and anthropometric status: metabolically unhealthy obesity (MUO), metabolically unhealthy normal weight, metabolically healthy obesity (MHO), and metabolically healthy normal weight (MHNW) [9]. Individuals with MHO have been shown to have a lower risk of CVD and mortality compared with those with MUO [11,12]. However, it is unclear whether MHO negatively affects health [13]—an uncertainty that is compounded by the lack of consensus regarding the definition of MHO [14]. The concept of MHO was first introduced in 2001, when it was reported that some individuals with obesity do not have any outcomes of poor health [15]. Obesity is usually defined by body mass index (BMI), waist circumference (WC), or body fat. The term MHO is used to describe obesity in which insulin sensitivity [16], blood pressure, glucose level, and lipid profiles [17] are normal, and there is no diagnosis of metabolic syndrome based on the criteria of the National Cholesterol Education Program, Adult Treatment Panel III (ATP III) [18], International Diabetes Federation (IDF) [19], Joint Interim Statement (JIS) Harmonized Criteria of the IDF [20], or other criteria [13,21]. The prevalence of MHO varies from 2% to 28% and is affected by metabolic criteria as well as sex, age, smoking, region, and alcohol consumption [22].

Two recent meta-analyses reported that, compared with participants with MHNW, those with MHO were at higher risk of cardiovascular events but not all-cause mortality [23,24]. In 2016, MHO was introduced as a Medical Subject Heading term to describe a metabolically “benign” obesity that is associated with a “risk” of CVD [25]. A meta-analysis was performed in 2019, focusing on the comparison of the four phenotypes of obesity [26]; however, limited articles discussing MHO were found by a limited keyword search and the results were not significant. To the best of our knowledge, there have been very few studies addressing the relationships between these phenotypes and risks for CVD or other morbidities with comprehensive evidence.

If a more coherent definition of MHO could be agreed upon, it could enable clarification of whether this phenotype is beneficial or harmful to individuals [27] Thus, the present study aimed to perform a comprehensive systematic review and meta-analysis to evaluate the relationship between MHO and CVD.

## 2. Methods

We conducted a systematic review and meta-analysis following a pre-established protocol registered on PROSPERO (CRD 42019130244), reported in accordance with PRISMA guidelines [28]. (Appendix A).

### 2.1. Definition of Metabolic Health and Outcomes

We extracted data relating to adults aged 18 years or older. Obesity was defined by BMI, WC, and body fat. Metabolic status was defined by insulin resistance, metabolic syndrome or metabolic disease diagnosed by blood glucose, blood pressure, or lipid profiles. We reported the outcomes for MHO compared with MHNW. The primary outcome was CVD as a composite of all fatal and nonfatal coronary heart disease (CHD), myocardial infarction (MI), stroke, heart failure (HF), and peripheral artery occlusion disease. The secondary outcome was all-cause mortality.

### 2.2. Data Sources and Search Strategy

In the present meta-analysis, we used comprehensive keywords to search large databases, adopted strict definitions, and utilized the PICO search tool. Two authors conducted the searches independently, and disagreements were resolved through discussion with the third author. Full search strategies are detailed in Appendix A. Briefly, we conducted electronic searches of the following databases, supplemented with hand-searching, from inception to May 2019: PubMed/Medline, EMBASE, Cumulative Index to Nursing and Allied Health Literature, and the Cochrane database. We used the keyword (MHO) to identify articles published after 2016 and keywords (obesity OR body mass index) AND (metabolic) AND (normal OR healthy OR benign) to identify earlier articles. Outcomes were identified using the keywords (CVD OR CHD OR MI OR stroke OR HF) AND (morbidity OR morbidities OR mortality OR incidence). We did not place constraints on language, year of publication, or participant characteristics (including participant age) in order to ensure a comprehensive search and identify articles that are aligned with our results of PICO hand-searches. Letters and editorials were excluded. We contacted authors to obtain additional information if necessary.

### 2.3. Study Selection and Methodological Quality Assessment

Inclusion criteria were studies on adults with obesity and normal metabolic status, studies that reported the outcome measures of interest as primary or secondary outcomes of the paper, and cohort studies. Exclusion criteria were duplicate publications, irrelevant articles, studies where MHO and metabolically healthy overweight were not clearly defined, studies that did not provide a comparison with individuals with MHNW, and articles reporting case series, cross-sectional studies, or reviews. We did not include data relating to outcome measures other than those stated above, such as transition to MUO or incidence of diabetes, metabolic syndrome, or other heart disease such as atrial fibrillation, diastolic dysfunction, myocardial function, subclinical atherosclerosis, and subclinical myocardial ischemia.

The Newcastle-Ottawa Scale evaluates the quality of nonrandomized studies by the quality of selection, comparability, and outcome [29]. After initial screening, two authors independently scored the selected studies using this scale. If the two authors disagreed, agreement was reached by consensus with the third author. Details of the scoring system are provided in Appendix A.

### 2.4. Data Extraction

Four authors independently extracted the following data: last name of the first author, year of publication, participants’ characteristics, definition of obesity, definition of metabolic health, variables that were adjusted, definition of outcomes, and major findings (Appendix A).

### 2.5. Statistical Analyses

For continuous outcomes, data were analyzed using the odds ratio (OR) with 95% confidence intervals (CIs). Study-level information are presented as medians with ranges. All analyses were carried out using R software version 1.1.456 [30]. Assuming that the true effect size was not the same, we employed a random-effects model using DerSimonian and Laird’s methods [31]. Results are presented in forest plots. Heterogeneity was quantified using the Cochran Q test and *I*^2^ statistics [32] and explained by prespecified subgroup analyses. For cumulative meta-analysis, the included studies were arranged in chronological order; then, multiple meta-analyses were performed by grouping studies by study year. We conducted a dose-response analysis to evaluate the linear relationships between BMI and the outcomes. We extracted data on BMI, number of participants, and person–years. The lowest boundary was assigned to the first BMI category (normal weight), as the reference group. The midpoint values of the BMI categories of overweight or obesity were used as the corresponding doses of outcomes. If the category had no upper boundary, the corresponding BMI was calculated as the lower boundary plus 1.5 times the range of the neighboring category. We estimated study-specific linear trends between BMI and the outcomes using a method developed by Greenland and Longnecker [33], then pooled the trends for random-effects meta-analysis. Weight-adjusted multivariate meta-regression models were used to test the contributions of effect modifiers (age, sex, follow-up duration, and smoking) [34]. Adjusted *R*^2^ is commonly used to quantify the goodness of fit of our model in percentage (0–100%). We assessed small-study effects using funnel plots and Egger’s test [35]. Sensitivity analyses were conducted by considering the quality of the included studies, omitting each study and excluding CV mortality from CV morbidity in turn to test the robustness of the results.

## 3. Results

### 3.1. Description of Studies and Quality Assessment

Figure 1 illustrates the search process. A total of 43 cohort studies were included; all included and excluded studies are listed in Appendix A. The characteristics of the included studies are shown in Appendix A. One article presented the hazard ratio of MHO and MHNW with metabolically healthy overweight as the reference; however, we could not obtain the correlation of MHO and MHNW after contacting the authors and thus excluded the study [36]. All of the included studies were published after 2004, and most were conducted in the United States or Europe. In total, 4,822,205 participants were included, with a median prevalence of MHO of 6.6% (range, 1.2–31.0%). The median participant age was 49.9 (30.3–74.0) years; the median proportion of women was 52.0% (0–100%); and the median smoking rate was 20% (5.7–67.6%). The median follow-up duration was 10.6 (1.0–30.0) years.

CDSR, Cochrane Database of Systematic Reviews; CINAHL, Cumulative Index to Nursing and Allied Health Literature; MHO, metabolically healthy obesity; MHOW, metabolically healthy overweight; MHNW, metabolically healthy normal weight.

The mean score (± standard deviation) of the included studies according to the Newcastle-Ottawa Scale was 7.9 ± 1.0, out of a possible score of 9 (Appendix A). Most of the included studies had a quality score higher than 7. A study published in 2004 had the lowest score of 5, as the study did not adjust for smoking, assessed CVD outcome by telephone or mail contact, patients were followed up for 3.5 years only, and the dropout rate was not reported [37]. Two studies were scored 6; one of which used self-reported BMI and did not report baseline CVD or dropout rate [38] and the other determined CVD using an epidemiological questionnaire with follow-up of only 3.2 years, with no report of dropout rate [39]. We did not exclude any of the articles with quality scores below 7, but we performed sensitivity analysis to establish any effects of their inclusion.

### 3.2. Results of the Meta-Analysis

To evaluate the primary outcome of CVD, 35 cohort studies were pooled for the meta-analysis. Five of our included studies reported CV mortality as their CVD endpoint, which was part of our secondary outcome, all-cause mortality which refers to death of any reason. The cumulative forest plot showed an increase in the risk of CVD since 2005 (Figure 2). Participants with MHO were at significantly higher risk of CVD than individuals with MHNW (Table 1). Because of the underlying heterogeneity of definitions and outcomes, we performed subgroup analysis. Some of the studies used modified criteria of metabolic syndromes, which we aggregated for the purposes of the present study. Compared with participants with MHNW, those with MHO were at significantly higher risk of CVD, as defined by the modified ATP III criteria, modified IDF criteria, insulin resistance, modified JIS criteria, and other definitions. The forest plot is shown in Appendix A.

Compared with participants with MHNW, those with MHO were at significantly higher risk of CVD when defined by BMI, but not when MHO was defined by WC (Table 1). The forest plot is shown in Appendix A. We used the composite outcome of CVD comprising all fatal and nonfatal CHD, MI, and HF. Compared with participants with MHNW, those with MHO were at significantly higher risk of CHD/MI, CVD mortality, and fatal and nonfatal CVD, but not HF (Table 1). The forest plot is shown in Appendix A.

A total of 11 articles with 35 BMI categories were pooled for dose-response analysis, which revealed a significant linear relationship between BMI and the risk of CVD (*p* < 0.001). For every unit increase in BMI, the risk of CVD increased by 2% (OR for slope, 1.019; Figure 3).

To explain the residual heterogeneity and to better understand the potential effect modifier, we performed prespecified meta-regression analyses of sex, age, follow-up duration and smoking status (Table 2). Univariate meta-regression showed that the risk of CVD due to MHO was borderline nonsignificant when modified by age, proportion of women, and smoking (bubble plots are shown in Figure 4 and Appendix A). Multivariate meta-regression model analysis showed that as the proportion of women and mean age increased, the impact of MHO on the risk of CVD diminished significantly. Smoking and follow-up duration did not modify the effect significantly (Appendix A). The proportion of heterogeneity explained by the meta-regression is represented by *R*^2^. Age, sex, and smoking accounted for 99.99% of heterogeneity in terms of MHO and the risk of CVD. The funnel plot showed no substantial asymmetry, and Egger’s test indicated no publication bias (*p* = 0.73; Appendix A). We excluded articles with Newcastle-Ottawa Scale scores below 7, omitted each study individually and excluded CV mortality from CV morbidity to perform sensitivity analyses. Overall, these statistics indicated that the results were robust (Appendix A).

Each bubble represents a study and bubble size represents the sample size of the study. The regression line shows a nonsignificant trend of declining risk with larger women proportion. OR = 0.77 (0.50; 1.19), *p* = 0.23, *R*^2^ (%) = 0%.

A total of 20 cohort studies were pooled to evaluate the secondary outcome of all-cause mortality (Table 1). The cumulative forest plot showed an increased risk of all-cause mortality since 2005 (Appendix A). Participants with MHO had significantly higher rates of all-cause mortality than participants with MHNW. The subgroup analysis showed that, compared with participants with MHNW, those with MHO that was defined by insulin resistance had significantly higher rates of all-cause mortality. Participants with MHO defined by the modified ATP III criteria had borderline increased all-cause mortality. The modified JIS criteria or other definitions did not give similar results for all-cause mortality (the forest plot is shown in Appendix A). The subgroup analysis revealed that participants with MHO, defined by either WC or BMI, were at a significantly higher risk for all-cause mortality (the forest plot is shown in Appendix A).

A total of 9 articles with 26 BMI categories were pooled for dose-response analysis, which revealed a nonsignificant linear relationship between BMI and the risk of all-cause mortality (OR for slope, 1.01; *p* = 0.29; Appendix A). The univariate meta-regression analysis revealed that age, smoking status and follow-up duration were borderline and nonsignificant effect modifiers (*p* = 0.07, *p* = 0.77 and *p* = 0.33, respectively; bubble plots are shown in Appendix A). Both univariate and multivariate meta-regression analyses indicated that as the proportion of women increased, the impact of MHO on the risk of all-cause mortality decreased significantly (*p* = 0.043 and *p* = 0.015, respectively; bubble plots are shown in Appendix A). Among sexes, the risk of all-cause mortality associated with MHO decreased significantly as age increased (*p* = 0.020). Age and sex accounted for 95.5% of heterogeneity in terms of MHO and the risk of all-cause mortality (Appendix A). The funnel plot was symmetrical, and Egger’s test indicated no significant publication bias (*p* = 0.62, Appendix A). We excluded articles with Newcastle-Ottawa Scale scores below 7 and omitted each study individually to perform sensitivity analyses. T results remained robust (Appendix A).

## 4. Discussion

Our analysis demonstrates that individuals with MHO are at significantly higher risk of CVD and all-cause mortality, with BMI, sex, age, and smoking habits influencing the risk in such individuals.

Our results are consistent with three previous meta-analyses that reported that MHO caused the risk of CVD to increase significantly by 45% to 100% [23,24,40] compared with MHNW. A recent meta-analysis included only 21 studies and did not find a significant risk of CVD associated with MHO [26]. All-cause mortality was only discussed in one study included in the present meta-analysis, but the association with MHO was not found to be significant [24]. Compared with previous meta-analyses, we identified more studies and reported a more accurate pooled effect size and more precise 95% CI. This can be attributed to our more up-to-dated criteria and comprehensive approach to database searches.

Despite using different definitions of MHO, the included studies consistently defined metabolic syndrome with the core concept of insulin resistance. In our study, we observed a higher risk of CVD when MHO was defined by insulin resistance than when other definitions were considered. The low heterogeneity that we observed may be attributed to consistency in the method of homeostasis model assessment (HOMA) among the studies. HOMA is a method used to quantify insulin resistance and beta cell function, computed as the product of fasting plasma glucose (FPG, mmol/L) and fasting serum insulin (mU/L) divided by 22.5. Lower values indicate high insulin sensitivity; whereas higher values indicate low insulin sensitivity or insulin resistance. Across our included studies, the cut-off value was based on a definite value or the quartile of the distribution among cohorts. The ATP III system [18] was found to be the most widely used, which defines metabolic syndrome as the presence of any three of the following traits: abdominal obesity defined as WC ≥102 cm in men or ≥88 cm in women; serum triglycerides ≥150 mg/dL or drug treatment for elevated triglycerides; serum high-density lipoprotein cholesterol (HDL-C) <40 mg/dL (1.03 mmol/L) in men or <50 mg/dL in women or drug treatment for low HDL; blood pressure ≥130/85 mmHg or drug treatment for elevated blood pressure; FPG ≥100 mg/dL or drug treatment for elevated blood glucose [41]. The IDF updated their metabolic syndrome criteria in 2006. Central obesity with ethnicity-specific WC cutoff points is an essential criterion, plus any two of the four following traits: triglycerides ≥150 mg/dL or treatment for elevated triglycerides; HDL-C <40 mg/dL in men or <50 mg/dL in women or treatment for low HDL; systolic blood pressure ≥130 mmHg, diastolic blood pressure ≥85 mmHg or treatment for hypertension; FPG ≥100 mg/dL or previously diagnosed type 2 diabetes (an oral glucose tolerance test is recommended, but not required, for patients with elevated FPG) [19]. The latest JIS definition was proposed in 2009 and includes the same variables as the IDF criteria, although central obesity is not an essential component [20]. The studies included in the present analysis were found to use modified versions of the above criteria by using substituted or adjusted cutoff values of WC or FPG (or no cutoffs). Some studies used different definitions of metabolic syndrome, such as biomarkers including high-sensitivity C-reactive protein or apolipoprotein B, [42] whereas some defined the condition by diagnosis or treatment for hypertension, dyslipidemia, or diabetes. These modifications to the criteria demonstrated the inconsistent definitions of MHO, which could be part of the within-study variance. Furthermore, one article reported the risk of all-cause mortality to be lower in patients with chronic kidney disease with MHO [43]; the underlying clinical differences may have contributed to this discrepancy with our results and the heterogeneity of our analysis. A single study showed a decreased risk of HF in participants with MHO, although the small sample size of only 550 participants may mean that firm conclusions cannot be drawn [44].

The prevalence of MHO in our study was consistent with a previous report of an overall incidence of 7.3% [17]. None of our included studies reported the sex-specific impact at the individual-level. We found age and sex to be significant effect modifiers in the study-level. Although sex has been reported to play a role in the developmental programming of metabolism, [45] there are insufficient studies supporting our finding of the influence of sex. However, one study that focused on the outcome of hypertension showed a sex-specific impact of MHO [46]. Subgroup analyses performed as part of previous meta-analyses were underpowered to differentiate the modifying effects of sex, [26] smoking status, [24] and age [24]. The negative slope of age revealed that young participants with MHO had more risk of CVD than elderly participants with MHO, which may be related to sarcopenia and underlying diseases in the elderly. Early intervention of weight reduction is encouraged. More studies focusing on effect modifiers for the outcome of CVD in MHO are warranted.

The pathophysiology of MHO is considered as subclinical adipose tissue inflammation that often results in insulin resistance and is measured by predictors including C-reactive protein, interleukin 6, and free fatty acid levels; transition of adipose tissue leading to a metabolic state; and regulatory genetic predisposition involving the processes of apoptosis, adipogenesis, angiogenesis, and dysregulation of epigenetic adaptation hypothesis [47]. Metabolic support with D-ribose, coenzyme Q10, L-carnitine, and magnesium can improve the maintenance of contractile reserve and energy charge in minimally oxidative ischemic or hypoxic heart tissues [48]. Indirect evidence has revealed that supplementation with these nutrients may be helpful in maintaining heart function and weight loss [49,50]. Metabolic dysfunction in heart tissues may also explain the mechanism underlying the risk of CVD in MHO.

All phenotypes of obesity are considered to represent a state of disease. Individuals with MHO should not be considered as healthy, but as being in a “pre-metabolic syndrome” state [51] and at risk of future metabolic dysregulation or obesity-related health consequences. The concept of MHO seems most relevant in individuals who have mild, or class I, obesity (defined as BMI of 30 to <35) [17]. Moderate weight loss can reduce many unfavorable physiologic changes that are associated with obesity, and the cardiometabolic risks can be lowered. Therefore, every individual with obesity should be encouraged to achieve a normal weight in the long term [21].

The strength or the present study is that we employed a comprehensive search with strict definitions. Our analysis included millions of participants with long-term follow-up data from good-quality studies. However, our study does have several limitations that should be acknowledged. First, no randomized controlled studies were available and a causal relationship could not be established. We limited our analysis to cohort studies in order to establish temporality. Confounding factors in each study were adjusted using different models. The effect size of our results was moderate without publication bias, which is consistent with previous studies. Furthermore, our results indicate a plausible mechanism and coherence with epidemiological findings, as dose-response meta-analysis showed a significant linear association between BMI and the risk of CVD. Second, although our study evaluated obesity defined by various criteria, most of the included studies evaluated obesity by BMI only. Recent publications have suggested that the use of BMI alone is not a suitable indicator of obesity [52,53]. More research using indicators other than BMI is required. Third, the relationship between metabolically healthy underweight status and CVD is unknown; there is insufficient evidence to ascertain whether a J-shaped relationship exists between weight and CVD risk. Fourth, our included populations were of different ethnicities and from different centers. This diversity may not only be related to different definitions of obesity and metabolic health, but also related to the different CVD outcomes.

## 5. Conclusions

We provide robust evidence of a significant association of MHO with increased risk of CVD. Every incremental increase in BMI linearly increases this risk further. Therefore, all individuals with obesity should be encouraged to achieve a normal weight as early as possible and weight gain should also be discouraged in non-obese individuals.

## Figures and Tables

**Figure 1 jcm-08-01228-f001:**
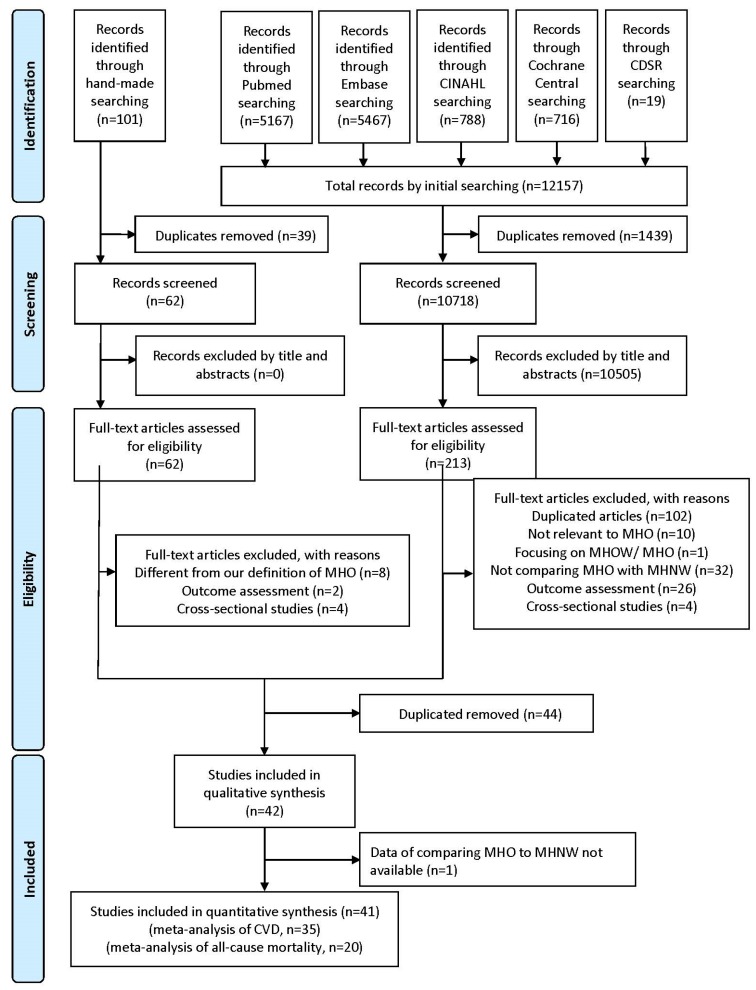
Flowchart of the study selection process.

**Figure 2 jcm-08-01228-f002:**
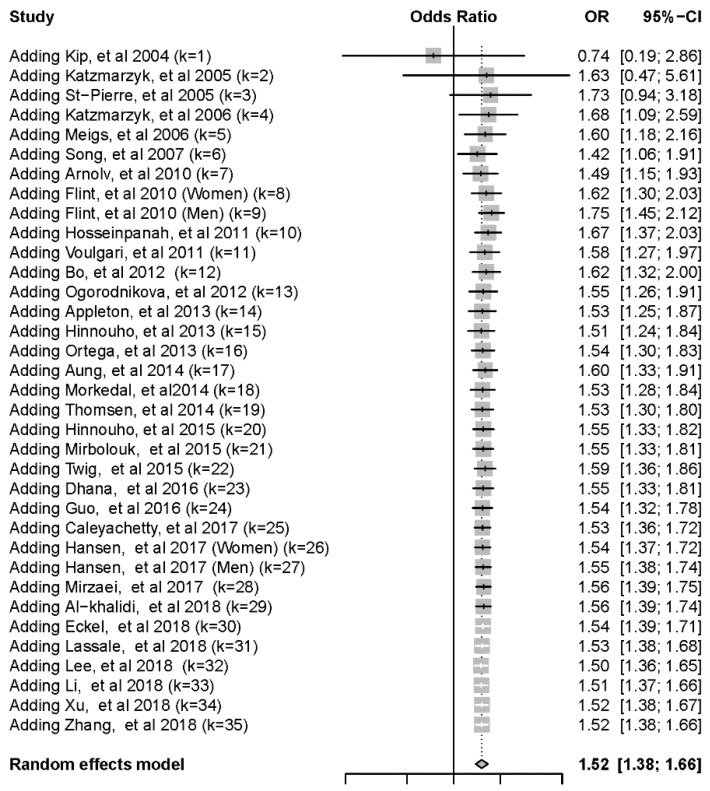
Cumulative forest plot of risk of cardiovascular disease from all included studies. CI, confidence interval; OR, odds ratio.

**Figure 3 jcm-08-01228-f003:**
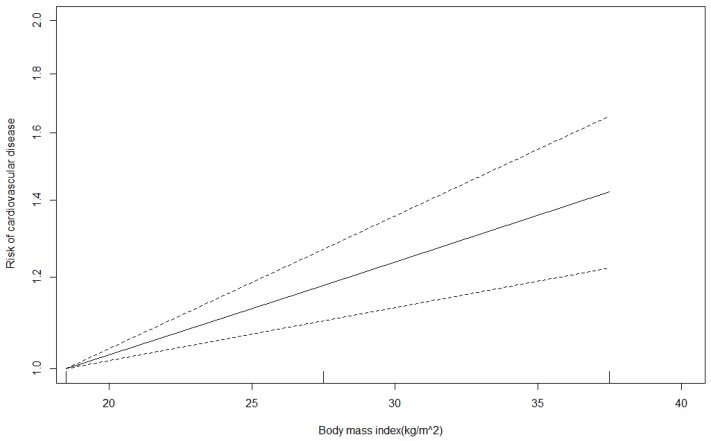
Dose-response analysis of body mass index and the risk of cardiovascular disease.

**Figure 4 jcm-08-01228-f004:**
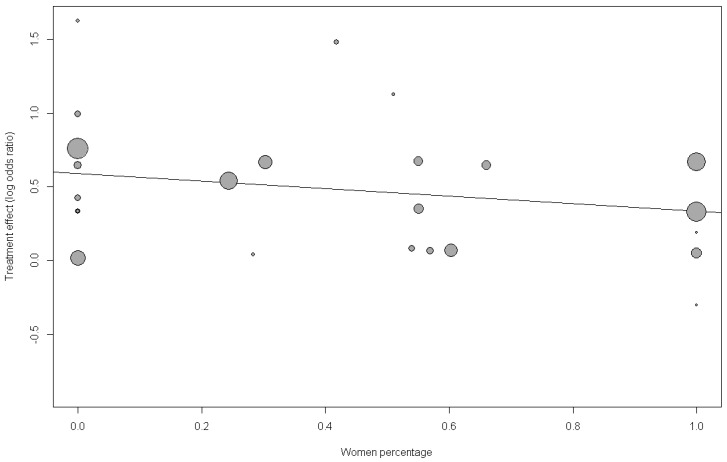
Meta-regression bubble plot of correlation between log odds ratio of cardiovascular disease and the proportion of women.

**Table 1 jcm-08-01228-t001:** Risk of Cardiovascular Disease and All-cause Mortality Associated with Metabolically Healthy Obesity in Different Subgroup Analyses.

	Risk of Cardiovascular Disease				All-Cause Mortality			
	OR (95% CI)	*p* Value for between-Group Differences	*I*^2^ (%)	Number of Studies	OR (95% CI)	*p* Value for between-Group Differences	*I*^2^ (%)	Number of Studies
Overall	1.52 (1.38; 1.66)		61	35	1.23 (1.05; 1.43)		62	20
Subgroups								
Definitions of metabolic health		0.17				0.32		
Modified ATP-III	1.43 (1.10; 1.85)		63	11	1.29 (1.00; 1.66)		0	5
Modified IDF	1.30 (1.06; 1.60)		53	5	NA		NA	NA
Insulin resistance	1.72 (1.30; 2.26)		0	5	1.56 (1.19; 2.05)		12	5
Modified JIS	1.45 (1.23; 1.72)		19	5	1.12 (0.78; 1.62)		77	4
Others	1.79 (1.49; 2.16)		77	9	1.10 (0.81; 1.49)		76	6
Definitions of obesity		0.83				0.27		
Body mass index	1.52 (1.38; 1.67)		62	34	1.20 (1.02; 1.43)		64	18
Waist circumference	1.40 (0.65; 2.98)		NA	1	1.45 (1.09; 1.94)		0	2
Different outcomes		0.14						
Coronary heart disease/myocardial infarction	1.39 (1.17; 1.65)		49	7				
CVD mortality	1.64 (1.12; 2.39)		0	5				
Fatal and nonfatal CVD	1.57 (1.40; 1.77)		68	22				
Heart failure	0.41 (0.11; 1.48)		NA	1				

ATP III, National Cholesterol Education Program, Adult Treatment Panel III; CI, confidence interval; CVD, cardiovascular disease; IDF, International Diabetes Federation; JIS, joint interim statement; NA, not applicable; OR, odds ratio.

**Table 2 jcm-08-01228-t002:** Meta-regression Analysis of Association between Covariates and Risk of Cardiovascular Disease.

	Risk of Cardiovascular Disease							
	Univariate				Multivariate			
	Unadjusted OR (95% CI)	*p* Value	*I*^2^ (%)	*R*^2^ (%)	Adjusted OR (95% CI)	*p* Value	*I*^2^ (%)	*R*^2^ (%)
Sex								
Men	1 (Reference)	NA			1 (Reference)	NA		
Women	0.77 (0.50; 1.19)	0.23	59.3	0	0.65 (0.46; 0.90)	0.014	59.3	0
Age, per year increase	0.99 (0.97; 1.00)	0.06	60.6	1.6	0.98 (0.97; 1.00)	0.030	39.1 ^1^	28.5 ^1^
Smoke status								
Nonsmoker	1 (Reference)	NA			1 (Reference)	NA		
Smoker	1.00 (0.99; 1.01)	0.94	65.5	0	1.01 (1.00; 1.02)	0.17	0 ^2^	99.99 ^2^
Follow-up duration, per year increase	1.01 (0.99; 1.03)	0.43	64.5	2.2	1.00 (0.97; 1.02)	0.69	41.1 ^3^	18.1 ^3^

CI, confidence interval; NA, not applicable; OR, odds ratio. Bold font represents statistically significant results. ^1^ Covariates: female sex and age. ^2^ Covariates: female sex, age, and smoker ^3^ Covariates: female sex, age and follow-up duration.

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
