# Peer review of "The Relationship between Metabolically Healthy Obesity and the Risk of Cardiovascular Disease: A Systematic Review and Meta-Analysis"

_jcm, 2019, doi:10.3390/jcm8081228_

Round 1
Reviewer 1 Report
excellent analysis. Suggest to make measures of insulin sensitivity/ resistace more clearly defined and how was it presented in the studies
2) can you obtain information on durations of patient's follow ups in the studies and to what extend follow ups may bias results; that is more events would have happened if longer durations were observed
Please add a generic statement on limitations due to differing ethnicities and center effect.
Author Response
Dear reviewer, please kindly see the attached file.

Reviewer 2 Report
Review for JCM-563086: the relationship between metabolically healthy obesity and the risk of cardiovascular disease: a systematic review and meta-analysis.
This study summarizes existing evidence on the association between metabolically healthy obesity and risk of cardiovascular disease, all-cause mortality as well as association between body mass index and risk of cardiovascular disease in metabolically healthy individuals. While comprehensive, interesting and well-written, there are some issues which need to be addressed to further improve the manuscript.
Major comments:
The authors should include a sensitivity analysis excluding the 5 studies on CVD mortality from the overall CVD estimates. It is a bit ambiguous to lump CVD morbidity and mortality together. Since the authors found sex to be a significant determinant of CVD morbidity and all-cause mortality estimates, they should then report sex-specific estimates for both outcomes, for completeness. What are the components of all-cause mortality reported across studies? Were the 5 studies on CVD mortality also part of the all-cause mortality estimates? A brief summary should be presented in the beginning of the results section.Minor comments
Change dose-response trend to linear trend all through the manuscript Lines 31-32 should read “…between BMI and CVD risk among the metabolically healthy” Lines 33-35: state that these variables explained the observed heterogeneity and report the adjusted R2 Lines 64-67: It is unclear how this 2019 meta-analysis differs from the present work under review. Please add more details to justify present work under review. Lines 117-118: This is wrong. Categorical variables cannot be summarized by medians. Please correct Table 1: Please add the p-values of the subgroup differences to Table 1. Line 133: Add the adjusted R2 In the lowest box of Figure 1, please specify the number of articles that was included for each of the three study objectives Lines 271-272: The modifications to the criteria do not explain the observed heterogeneity. Age, sex and smoking already explained 99.99% of the variance, and all subgroup estimates were positive and significant. The p-values from Table 1 should be non-significant for this variable. Please clarify. Line 282: change “interaction” to “modifying” Figure 3 title should be on BMI and CVD risk. Please edit Lines 245-248: Please rephrase as other studies were also accurate. I suggest to use “more up to date”. In addition, applying strict criteria does not yield more results. Please rephrase these lines. The importance of the linear trend observed between BMI and CVD risk should be reflected in the conclusion. Weight gain should also be discouraged in the non-obese given this finding.
Author Response

(The authors gave the same response as above.)
